

# A systematic review of the application of machine learning in the detection and classification of transposable elements

Simon Orozco-Arias[1,2], Gustavo Isaza[2], Romain Guyot[3,4] and Reinel Tabares-Soto[4]

[1] Department of Computer Science, Universidad Autónoma de Manizales, Manizales, Caldas, Colombia
[2] Department of Systems and Informatics, Universidad de Caldas, Manizales, Caldas, Colombia
[3] Institut de Recherche pour le Développement, CIRAD, University of Montpellier, Montpellier, France
[4] Department of Electronics and Automation, Universidad Autónoma de Manizales, Manizales, Caldas, Colombia

Corresponding author
Simon Orozco-Arias,
simon.orozco.arias@gmail.com

## ABSTRACT

**Background:** Transposable elements (TEs) constitute the most common repeated sequences in eukaryotic genomes. Recent studies demonstrated their deep impact on species diversity, adaptation to the environment and diseases. Although there are many conventional bioinformatics algorithms for detecting and classifying TEs, none have achieved reliable results on different types of TEs. Machine learning (ML) techniques can automatically extract hidden patterns and novel information from labeled or non-labeled data and have been applied to solving several scientific problems.

**Methodology:** We followed the Systematic Literature Review (SLR) process, applying the six stages of the review protocol from it, but added a previous stage, which aims to detect the need for a review. Then search equations were formulated and executed in several literature databases. Relevant publications were scanned and used to extract evidence to answer research questions.

**Results:** Several ML approaches have already been tested on other bioinformatics problems with promising results, yet there are few algorithms and architectures available in literature focused specifically on TEs, despite representing the majority of the nuclear DNA of many organisms. Only 35 articles were found and categorized as relevant in TE or related fields.

**Conclusions:** ML is a powerful tool that can be used to address many problems. Although ML techniques have been used widely in other biological tasks, their utilization in TE analyses is still limited. Following the SLR, it was possible to notice that the use of ML for TE analyses (detection and classification) is an open problem, and this new field of research is growing in interest.

## INTRODUCTION

Transposable elements (TEs) are genomic units with the ability to move from one locus to another within the genome. TEs have been found in all organisms and comprise the majority of the nuclear DNA content of plant genomes (*Orozco-Arias et al., 2018*), such as in wheat, barley and maize. In these species, up to 85% of the sequenced DNA is classified into repeated sequences (*Choulet et al., 2014*), of which TEs represent the most abundant and functionally relevant type (*Ventola et al., 2017*). Due to the high diversity of TE structures and transposition mechanisms, there are still numerous classification problems and debates on the classification systems (*Piégu et al., 2015*). TEs in eukaryotes are traditionally classified based on if the reverse transcription is needed for transposition (Class I or retrotransposons) or not (Class II or DNA transposons) (*Schietgat et al., 2018*). Retrotransposons can be further subclassified into four orders according to structural features and the life cycle of the element.

In plants, Long Terminal Repeat retrotransposon (LTR-RT) is the most frequent order (*Gao et al., 2012*; *Grandbastien, 2015*) and can contribute up to 80% of the plant genome size (e.g., in wheat, barley or the rubber tree) (*Rahman et al., 2013*). However, in humans, the non-LTR-RT order is the most common and is related to diseases such as cancer (*Tang et al., 2017*). Other levels of classification of TEs include sub-classes (only for DNA transposons which are distinguished by the number of DNA strands that are cut during transposition (*Wicker et al., 2007*)), superfamilies, lineages, and families (*De Castro Nunes et al., 2018*; *Neumann et al., 2019*).

Although several methods have been developed to detect TEs in genomes, including de novo, structure-based, comparative genomic and homology-based (reviewed in (*Orozco-Arias, Isaza & Guyot, 2019*)), there is no single bioinformatics tool achieving reliable results on different types of TEs (*Loureiro et al., 2013b*). Most of the algorithms available currently use a homology-based approach (*Nakano et al., 2018b*), but this method can present limited potential due to the vast diversity at the nucleotide level of TEs. Also, the repetitive nature of TEs, as well as their structural polymorphism, species specificity, and high divergence rate even among close relative species (*Mustafin & Khusnutdinova, 2018*), represent significant obstacles and challenges for their analysis (*Ou, Chen & Jiang, 2018*). Despite the complexity, a well-curated detection and classification of TEs is important, due to these elements have key roles into genomes, such as in the chromosomal structure, their interaction with genes, and adaptation and evolution processes (*Orozco-Arias, Isaza & Guyot, 2019*) and their annotation could provide insights into genomic dynamics (*Wheeler et al., 2012*).

In recent years, machine learning (ML) has been used by life scientists as a system for knowledge discovery (*Ma, Zhang & Wang, 2014*), achieving promising results. ML can be defined as the process of designing a model that will be calibrated from the training information and a loss function through an optimization algorithm (*Mjolsness & DeCoste, 2001*). Based on these extracted patterns, algorithms can then predict results from unknown data. Main ML training methods can be classified into supervised learning and unsupervised learning (*Ceballos et al., 2019*). The goal of supervised learning is to predict

a discrete (classification) or continuous (regression) value for each data point by using a provided set of labeled training examples. In unsupervised learning, which is based on clustering algorithms, the goal is to self-learn inherent patterns within the data (*Zou et al., 2018*). The main objective of ML tasks is to optimize a cost function in terms of a set of parameters for a proposed model. In the optimization process, the proposed model is calibrated. With this aim, the data are randomly split into a minimum of three subsets (named hold-out method): training, validation, and test sets leaving the first set for learning patterns and hyper-parameters, the second set for choosing the best models, and the last set for obtaining more realistic accuracy. On the other hand, k-fold cross-validation randomly splits data into k-folds and then applies hold-out to each subset (*Zou et al., 2018*). This process is crucial to avoid overfitting (also called overtraining) or underfitting (undertraining), which both lead to poor predictive performances. Therefore, the algorithm must reach an appropriate balance between model flexibility and the amount of training data. An overly simple model will underfit and fail to let the data "speak" while an overly flexible model will overfit to spurious patterns in the training data and fail to generalize (*Zou et al., 2018*).

The design and implementation of a ML system is a complex process that can be done in three steps: (1) raw data preprocessing (i.e., features selection and extraction, data imputation, etc.), (2) learning or training of the model by using an appropriate ML algorithm or architecture (to calibrate the model) and (3) model evaluation through metrics (*Ma, Zhang & Wang, 2014*). In some cases, the preprocessing step is very complex and relies on complicated algorithms to automate this task or on experts in the field. The use of deep learning (DL) in ML addresses the issue of selecting the correct data representation or the best features (*Eraslan et al., 2019*), avoiding the need for an expert in the area. DL has evolved as a sub-discipline of ML through the development of deep artificial neural networks (DNN) (i.e., neural networks with many hidden layers), such as auto-encoders, fully connected neural networks (FNN), convolutional neural networks (CNNs), recurrent neural networks (RNNs), among others (*Eraslan et al., 2019*). DL has shown successful results in life sciences (*Yu, Yu & Pan, 2017*), especially in genomics. In this area, it has been used for identifying different types of genomic elements, such as exons, introns, promoters, enhancers, positioned nucleosomes, splice sites, untranslated regions, etc. (*Yue & Wang, 2018*).

Here, we performed a systematic review of applications of ML algorithms and architectures in TE detection and classification problems. We also discuss other uses of ML and DL in similar tasks that can be extrapolated to TE issues. To our knowledge, this is the first review focused mainly on the use of ML in TEs.

## SURVEY METHODOLOGY

We conducted an exhaustive literature review by applying the Systematic Literature Review (SLR) process proposed by *Kitchenham & Charters (2007)* and preferred reporting items for systematic reviews and meta-analyses guidelines (*Moher et al., 2009*) (Fig. 1). We followed the six stages of the review protocol used in *Wen et al. (2012)*
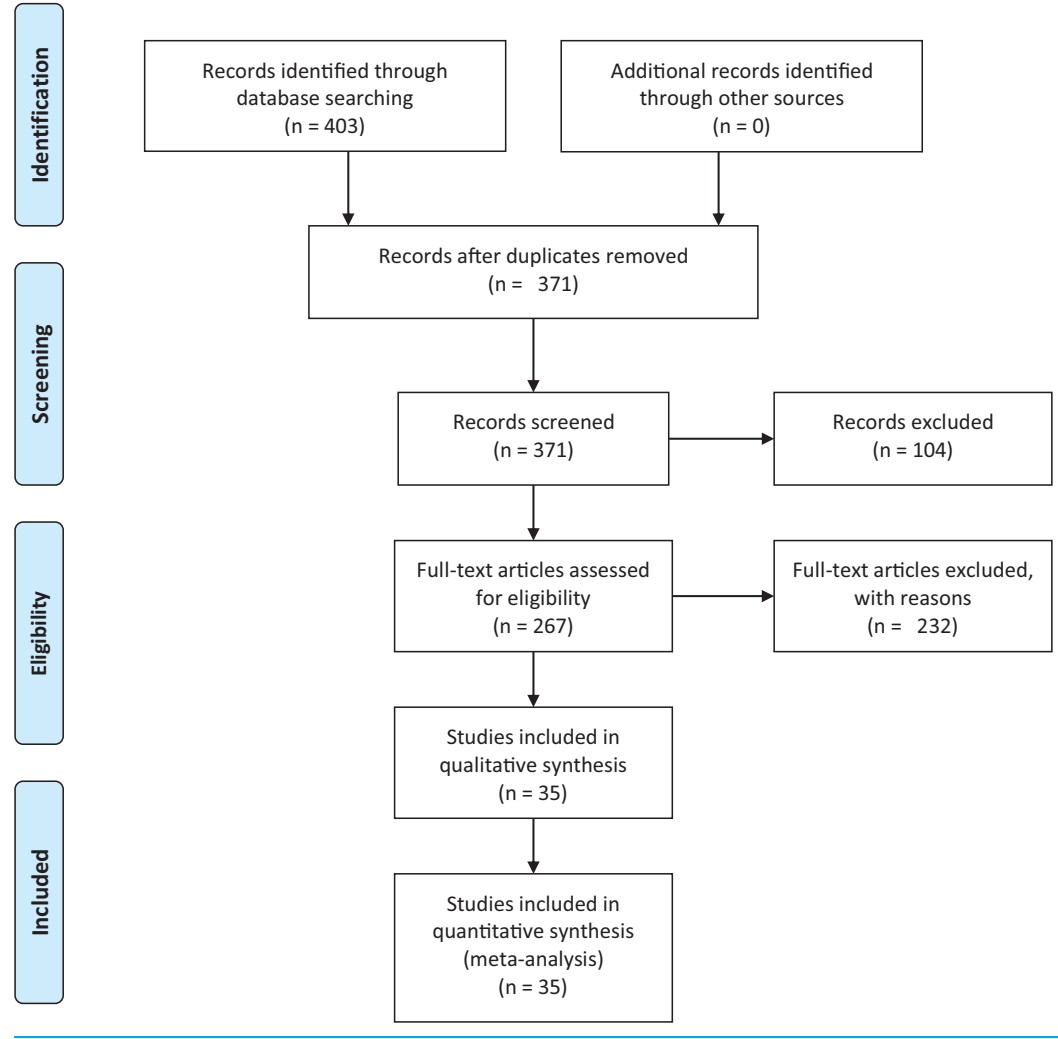

**Figure 1 PRISMA flow diagram.** PRISMA flow chart for search and article screening process. From: *Moher et al. (2009)*.

but added a previous stage, which aims to detect the need for a review (Fig. 2). First, we searched for other reviews to formulate questions related to the aim of this review, and then we selected the search strategy based on key terms and available databases. In the next step, we defined the exclusion criteria for filtering relevant articles that could contribute to answering the questions from the first stage. For this, we applied several filters in the Quality Assessment Checklist (fourth stage) to choose articles to be included in the following steps. Finally, we performed data extraction and data synthesis to process the information retrieved.

## Identification of the need for a review

The strategy used was based on the guidelines proposed by *Wen et al. (2012)* and used in *Franco-Bedoya et al. (2017)* and *Reinel, Raul & Gustavo (2019)*. To define the need for a systematic review, we searched through published and available reviews (secondary

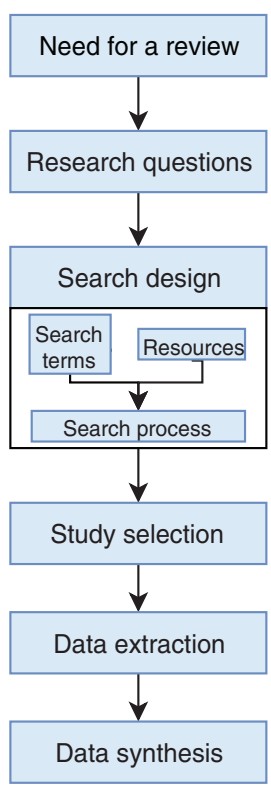

**Figure 2 Stages of the systematic literature review process.** Based on *Wen et al. (2012)*.

**Table 1 Literature resources used in this review.**

| Database | Link |
| --- | --- |
| Scopus | https://www.scopus.com |
| Science direct | https://www.sciencedirect.com/ |
| Web of science | https://clarivate.com/products/web-of-science/ |
| Springer link | https://link.springer.com/ |
| PubMed | https://www.ncbi.nlm.nih.gov/pubmed/ |
| Nature | https://www.nature.com/ |

studies) on the topic of interest. We used Eq. (1) to search in the literature databases referenced in Table 1.

$$\text{(``transposable element'' OR retrotransposon OR transposon) AND (``machine}$$
$$\text{learning'' OR ``deep learning'') AND (review OR ``systematic review'' OR overview} \quad (1)$$
$$\text{OR ``state of the art'' OR ``systematic mapping'')}$$

The keywords were selected based on the following: (i) type of information that we aimed to retrieve (TEs as well as retrotransposons and transposons classes), (ii) techniques addressed in this review (ML and DL) and (iii) keywords related to secondary studies.

All of the databases showed results using the search equation (106 results after filtering), yet only one secondary study (*Dashti & Masoudi-Nejad, 2010*) specifically applied ML

in TEs. However, this review published in 2010 focused only on support vector machine (SVM). Given the lack of secondary studies about this topic, we concluded that a systematic review of ML applications in TEs was needed.

## Research questions

The main aim of this review is to summarize and clarify trends, metrics, benefits, and possible ML techniques and architectures that have not yet been addressed in the detection and classification (for a graphical representation of the TEs classification, see Fig. 3) of TEs. With this in mind, we formulated the following questions:

1. Q1: Are ML approaches for TE analyses advantageous compared to bioinformatics approaches? It is important to identify if the application of novel tools like ML can contribute to improving current bioinformatics software. This is relevant since it is well-know that current methodologies are still far from yielding confident results in the detection and classification of TEs given their high variability and complexity (*Bousios et al., 2012*). On the other hand, many articles propose that TEs are involved in key characteristics of genomes, such as chromosome structure, environmental adaptation, and interspecific variability, among others. Therefore, the objective of Q1 is to determine how TEs detection and classification can be improved using ML to understand the dynamics and impacts of these elements better.

2. Q2: Which ML techniques are currently used to detect and classify TEs or other genomic data? We were interested in knowing which algorithms or architectures have been tested on TEs or other genomic data, such as long non-coding regions or retrovirus.

3. Q3: What are the best parameters and most used metrics in algorithms and architectures to detect and classify TEs? To avoid overfitting, it is important to use a splitting method to reduce dependance on the training data and to determine which type of data is better to use. Thus, we were interested in knowing which current articles addressed this step. Additionally, to compare algorithms and architectures, it is important to define metrics that accurately measure performance. It is also essential to assess if these techniques improve results compared to traditional bioinformatics software.

4. Q4: What are the most used DNA coding schemes in ML tasks? Because TEs comprise categorical data (nucleotides), there are many ways to transform this information into numerical data required by ML algorithms. Therefore, we were interested in understanding how this transformation can contribute to improving results and which coding schemes are widely used in this kind of problem.

## Search design and study selection

Once we identified the need for a review and formulated the research questions, we developed the search strategy to find research articles, chapter books, conference proceedings and other review articles in the databases shown in Table 1. Similar to search Eq. (1), we used general keywords related to (a) type of genomic data and (b) techniques used. We did not use specific keywords (such as specific algorithms or architectures) and any time limitations, because few relevant results were found on this topic. Major
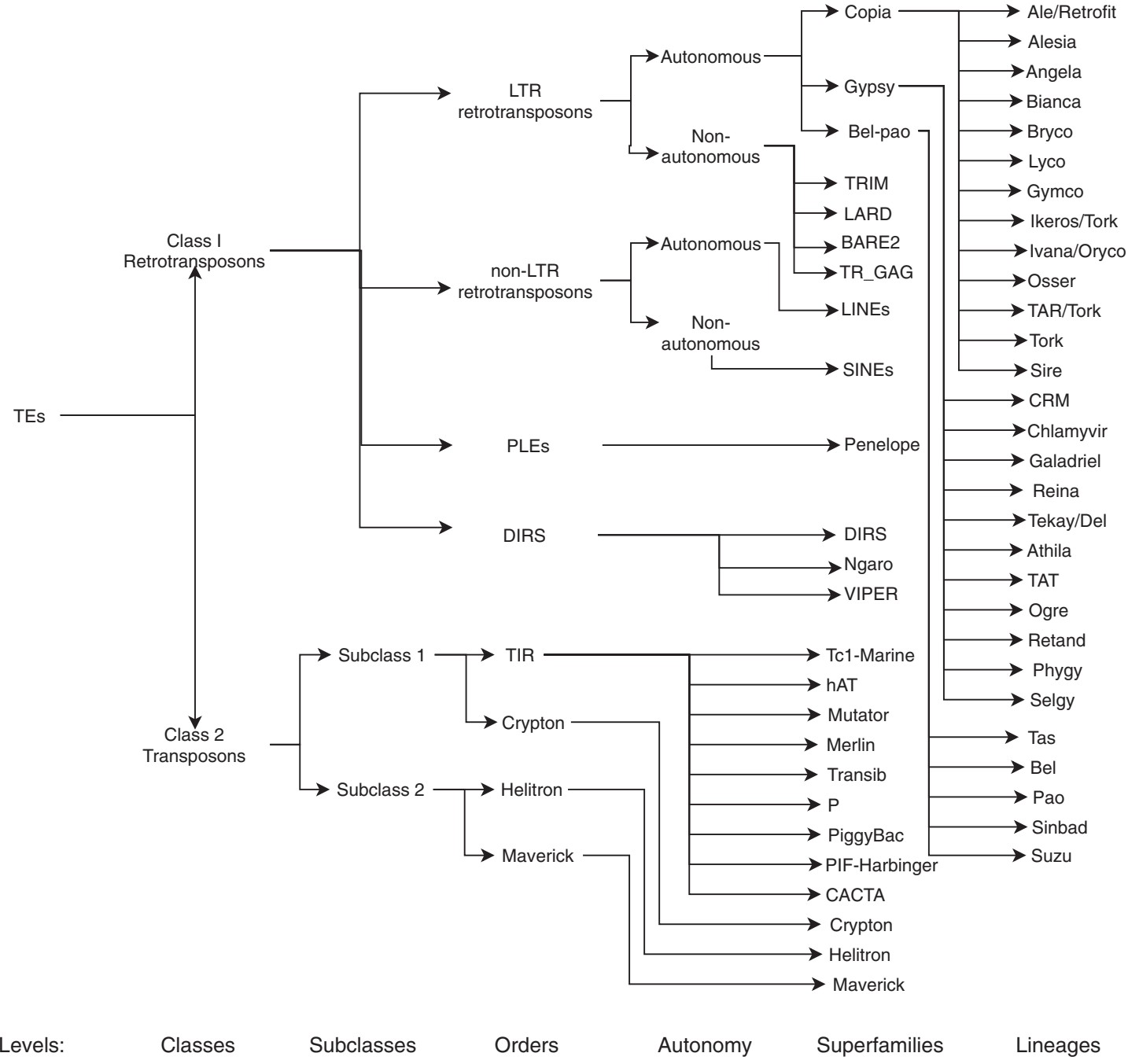

**Figure 3 Classification of TEs following Rexdb and GyDB nomenclatures.** Adapted from: *Schietgat et al. (2018).*

keywords were separated by the "AND" operator, and related keywords were linked using the Boolean operator "OR", as shown in Eq. (2):

("*transposable element*" OR *retrotransposon* OR *transposon*) *AND* ("*machine learning*" OR "*deep learning*")       (2)

The literature search using Eq. (2) retrieved 403 candidate articles of which were eliminated those that were: (a) repeated (the same study was found in different databases); (b) of different types (books, posters, short articles, letters and abstracts); (c) written in other languages (languages other than English). Then, we performed a fast read process (i.e., title, abstract, and conclusion) to detect articles that could contribute to answering the research questions. For this, we established the following inclusion and exclusion criteria.

Inclusion criteria

- Application of ML or DL in the detection of TEs (any class)
- Application of ML or DL in the classification of TEs (any class)
- Description of DNA coding schemes
- Use of ML or DL on similar genomic data
- Comparison of bioinformatics algorithms to ML or DL techniques
- Application of metrics to evaluate ML or DL algorithms for TEs or similar data

Exclusion criteria

- Do not use any ML or DL techniques
- Studies focused only on in vivo processes
- Studies that do not integrate any of the topics addressed in this review

After this selection process (Fig. 1), we identified 35 relevant articles that were used to extract and summarize the information.

## Data extraction and synthesis

In this stage, we first wholly read the selected publications (Table 2) to extract information to answer the research questions. Then, we registered the article into a data extraction card proposed by *Wen et al. (2012)* with some adaptations to our study. The card contained information on publication identifier, year, publication name, related research questions, and the information itself.

In the final step, we synthesized all of the collected information and obtained evidence to answer the research questions. Interestingly, more than 50% of the selected studies were published between 2017 and 2019 (Fig. 4), demonstrating a growing interest in this topic in the last years.

We identified 35 relevant publications after the SLR (see Table 2). These articles were published between 2009 and 2019. Among them, 77% (27) were reported in journals, 17% (6) in conference proceedings and 6% (2) as book sections (Fig. 5A). The selected articles were published in 21 journals, of which ten focused on bioinformatics or computational biology, six on genomics or genetics, and 15 on other areas (Fig. 5B).

## Benefits of ML over bioinformatics (Q1)

There is much literature about applications of ML in bioinformatics (e.g., reviewed in *Larrañaga et al. (2006)*), showing improvements in many aspects such as genome annotation (*Arango-López et al., 2017*). In recent years, much bioinformatics software has
**Table 2 Selected publications and their contribution to each research question.**

| Publication identifier | Year | Q1 | Q2 | Q3 | Q4 | References | Publication identifier | Year | Q1 | Q2 | Q3 | Q4 | References |
|---|---|---|---|---|---|---|---|---|---|---|---|---|---|
| P1 | 2017 | X | X | X | X | Yu, Yu & Pan (2017) | P19 | 2013 | X | X | X |  | Loureiro et al. (2013b) |
| P2 | 2018 | X | X | X |  | Schietgat et al. (2018) | P20 | 2014 |  | X | X |  | Ma, Zhang & Wang (2014) |
| P3 | 2017 | X | X | X |  | Arango-López et al. (2017) | P21 | 2010 | X | X |  |  | Dashti & Masoudi-Nejad (2010) |
| P4 | 2013 | X | X | X |  | Loureiro et al. (2013a) | P22 | 2010 |  | X | X |  | Ding, Zhou & Guan (2010) |
| P5 | 2011 | X | X | X |  | Tsafnat et al. (2011) | P23 | 2019 |  |  |  | X | Jaiswal & Krishnamachari (2019) |
| P6 | 2018 | X | X | X |  | Zhang et al. (2018) | P24 | 2015 | X | X | X | X | Girgis (2015) |
| P7 | 2019 | X | X |  | X | Eraslan et al. (2019) | P25 | 2018 |  | X | X | X | Nakano et al. (2018a) |
| P8 | 2018 | X | X | X |  | Douville et al. (2018) | P26 | 2018 |  | X | X | X | Zamith Santos et al. (2018) |
| P9 | 2018 |  | X | X |  | Chen et al. (2018) | P27 | 2009 |  | X |  |  | Abrusan et al. (2009) |
| P10 | 2012 | X | X | X | X | Ashlock & Datta (2012) | P28 | 2019 | X | X |  |  | Su, Gu & Peterson (2019) |
| P11 | 2017 | X | X | X |  | Smith et al. (2017) | P29 | 2017 | X | X | X | X | Nakano et al. (2017) |
| P12 | 2014 | X | X | X | X | Kamath, De Jong & Shehu (2014) | P30 | 2014 |  | X | X | X | Brayet et al. (2014) |
| P13 | 2016 | X | X | X |  | Kim et al. (2016) | P31 | 2013 |  | X |  |  | Zamani et al. (2013) |
| P14 | 2018 | X | X | X |  | Segal et al. (2018) | P32 | 2019 |  | X |  |  | Hubbard et al. (2019) |
| P15 | 2011 | X | X | X |  | Rawal & Ramaswamy (2011) | P33 | 2014 |  | X | X |  | Ryvkin et al. (2014) |
| P16 | 2017 | X | X | X |  | Tang et al. (2017) | P34 | 2013 | X | X | X | X | Zhang et al. (2013) |
| P17 | 2017 | X | X | X |  | Ventola et al. (2017) | P35 | 2019 |  | X | X |  | Da Cruz et al. (2019) |
| P18 | 2018 |  | X | X | X | Nakano et al. (2018b) | | | | | | | |

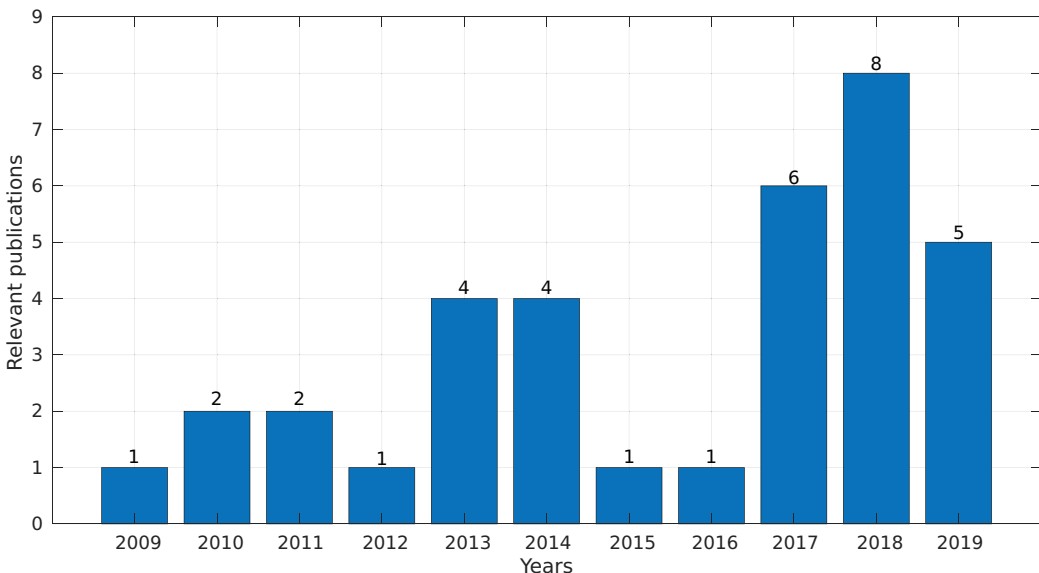

**Figure 4 Number of relevant publications found per year.**

been developed to detect TEs (*Girgis, 2015*) and, although they follow different strategies (such as homology-based, structure-based, de novo, and using comparative genomics), these lack sensitivity and specificity due to the polymorphic structures of TEs (*Su, Gu &*

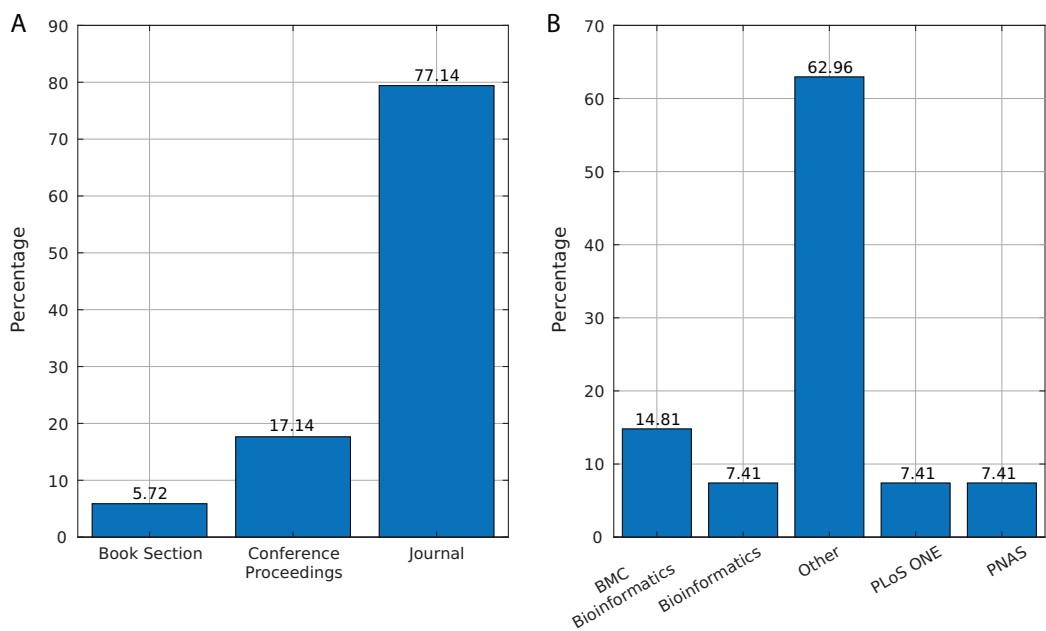

**Figure 5 Source of selected publications.** (A) Percentage of publications in each source. (B) Distribution of publications in journals.               

*Peterson, 2019*). *Loureiro et al. (2013a)* proved that ML could be used to improve the accuracy of TEs detection by combining results obtained by several conventional software and training a classifier using these results (*Schietgat et al., 2018*; *Loureiro et al., 2013b*). Loureiro's work provided novel evidence for the use of ML in TEs, yet it did not use ML to obtain the predictions, making the results too dependent on traditional algorithms. Using the Random Forest (RF) algorithm, *Schietgat et al. (2018)* were able to improve results obtained by popular bioinformatics software (which followed a homology-based strategy) such as Censor, RepeatMasker, and LTRDigest (*Schietgat et al., 2018*) in the detection of LTR retrotransposons. The authors proposed a framework called TE-Learner[LTR], which outperformed LTRDigest in recall and RepeatMasker and Censor in terms of precision.

Machine learning techniques also obtain better results than traditional methods regarding TEs classification. Using ML, it is possible to classify non-autonomous TEs (specifically derived from LTR retrotransposons) using features other than coding regions (which are commonly used in classification processes), including element length, LTR length, and ORF lengths (*Arango-López et al., 2017*). ML algorithms can distinguish between retroviral LTRs and SINEs (Short Interspersed Nuclear Elements) by combining a set of methods (*Ashlock & Datta, 2012*), which is a complicated procedure in bioinformatics. Also, using hierarchical classification, ML-based methods obtain better results than well-known homologous-based methods (specifically, BLASTn algorithm) (*Nakano et al., 2017*).

The advantages of using ML in bioinformatics include the discovery of entirely new information such as arrays of mobile genetic elements, new transposition unit boundaries (*Tsafnat et al., 2011*), and predicting new long noncoding RNA that are related to cancer

(*Zhang et al., 2018*). Other applications include extracting discriminatory features for automatically determining functional properties of biological sequences (*Kamath, De Jong & Shehu, 2014*), identifying DNA motifs, which is a difficult task in non-ML applications (*Dashti & Masoudi-Nejad, 2010*), and automating specific processes like the identification of long non-coding RNAs (*Ventola et al., 2017*) and the classification of LTR retrotransposons (*Arango-López et al., 2017*).

On the other hand, DL has been applied in biological areas such as genomics (for a review see, *Yue & Wang, 2018*) proving to be promising (*Yu, Yu & Pan, 2017*) due to the flexibility showed by deep neural networks. In *Eraslan et al. (2019)*, several applications in genomics are discussed such as variant calling, base calling for novel sequencing technologies, denoising ChIP–seq data (chromatin immunoprecipitation followed by sequencing), and enhancing Hi-C data resolution (Chromosome conformation capture followed by pair-end sequencing). Also, some frameworks allow users to use GPUs as a complement for CPUs, achieving a faster execution of DL algorithms (*Eraslan et al., 2019*). Deep neural networks have also been used to improve the prediction of global enhancers, which was proven to be challenging using other computational tools (*Kim et al., 2016*).

Machine learning and DL fields are supported by multiple companies and industry research groups, which anticipated the great benefits that artificial intelligence can contribute to genomics, human health (*Eraslan et al., 2019*), and major crops. Several articles using ML or DL techniques reported that TEs are associated with many human diseases (*Zhang et al., 2013*). For example, cancer-related long noncoding RNAs have higher SINE and LINE numbers than cancer-unrelated long noncoding RNAs (*Zhang et al., 2018*). Likewise, several types of epithelial cancers acquire somatic insertions of LINE-1 as they develop (mentioned in *Tang et al. (2017)*). Moreover, the genes that confer antibiotic resistance (called R genes) in bacteria are associated with TEs, and it is possible to detect them through ML (*Tsafnat et al., 2011*). Finally, although LTR retrotransposons are related to retroviruses such as HIV, ML algorithms can distinguish them from SINEs (*Ashlock & Datta, 2012*). On the other hand, the human genome is composed of a considerable number of interspersed repeats, such as LINE-1 as one of the most abundant (*Tang et al., 2017*)), human endogenous retroviral sequences comprising 8–10% of the genome, and SINEs contributing with ~11%. Meanwhile, protein-coding regions comprise only about 1.5% (*Ashlock & Datta, 2012*).

Since TEs are under relatively low selection pressure and evolve more rapidly than coding genes (*Rawal & Ramaswamy, 2011*), they undergo dynamic evolution. Moreover, insertions of other TEs (nested insertion), illegitimate and unequal recombination, cellular gene capture, and inter-chromosomal and tandem duplications (*Garbus et al., 2015*) make TEs classification and annotation a very complicated task (*Bousios et al., 2012*). Thus, conventional methods (such as bioinformatics) cannot obtain reliable results in TE detection and classification tasks.

In supervised problems, the process of feature extraction or selection is a crucial step for the performance of the entire architecture. In ML, the processes of selection of variables or characteristics must be carried out by a thematic expert. Deep network architectures, on

the other hand, allow characteristics to be extracted in a nonlinear and automatic way. The hidden layers of deep neural networks transform these characteristics into intricate patterns relevant to the classification problem (*Eraslan et al., 2019*). In the specific case of TEs, because they are DNA sequences, the extraction of characteristics is usually a too complex process due to a large amount of information, their unstructured form, and their sequentially. In this case, the deep neural networks provide new features that cannot be extracted manually. For example, CNNs can discover local patterns in sequential data such as pixels in an image or DNA (*Zou et al., 2018*). These patterns known as motifs have functions of great importance in the genome as promoters of genes and to be found in the LTR sequences of retrotransposons, in addition, if they are found in different places and under certain frequencies could be very useful to identify or classify TEs. Although motifs are essential for DNA classification problems, it is not enough to find the exact patterns, because DNA can undergo modifications or mutations and because specific motifs can function the same as others even if they do not have the same nucleotides. Another benefit of ML over bioinformatics is the use of labeled data to generate computational models. Currently, a few hundreds of plant genomes are available to train algorithms, but this number will increase significantly in the near future due to massive genome sequencing projects such as the 10 K plant project (https://db.cngb.org/10kp/) or the Earth BioGenome Project (https://www.earthbiogenome.org). This large amount of data will help to produce more accurate and reliable software thought ML and DL. Additional to available training data, some plant genomes are very interesting to identify and classify TEs following ML or DL approaches, due to their huge genome and their composition. As examples, sugarcane, maize, and barley have large genomes (3 Gb, 2 Gb and 5.1 Gb respectively) that are composed mainly by repetitive sequences (up to 80%, (*Rahman et al., 2013*)). On the other hand, the process of supervised training of ML algorithms provides another advantage when training a model that increases true positives (TP) and decreases false positives results. To improve this rate of performance, the hyperparameters of the models can be tuned using techniques such as search grid, which does an exhaustive search for the hyperparameters that produce the best accuracy and precision.

## ML architectures and algorithms currently used for TEs or similar data (Q2)

ML has been applied in bioinformatics due to a large amount of data that has been generated. *Ma, Zhang & Wang (2014)*, review the application of ML in topics such as genome assembly, genomic variation detection, genome-wide association studies, and the in silico annotation of coding genomic loci. Particular focus has been given to loci that code for proteins, TEs, noncoding RNAs, miRNAs and targets, transcription factor binding sites, cis-regulatory elements, enhancer and silencer elements, and mRNA alternative splicing sites. Additionally, many frameworks have been developed to facilitate the implementation of ML algorithms in bioinformatics projects. Tools such as Scikit-learn, Weka, and several packages developed in R (for a complete list, see, *Ma, Zhang & Wang, 2014*) allow using ML-based techniques in biological areas.

**Table 3 Machine learning algorithms used in publications selected in this study.**

| Publication | Data source | Task | ML algorithm | Learning method | References |
|---|---|---|---|---|---|
| P2 | Numerical and categorical features based on coding regions | Detect LTR Retrotransposons at the super-family level | RF | Supervised | *Schietgat et al. (2018)* |
| P3 | Numerical and categorical features | Classify LTR Retrotransposons at the lineage level | DT, BN and lazy algorithms | Supervised | *Arango-López et al. (2017)* |
| P4 | Numerical and categorical features | Improve the detection and classification of TEs | NN, BN, RF, DT | Supervised | *Loureiro et al. (2013a)* |
| P5 | Numerical features based on structure | Detect boundary sequences of mobile elements | HMM, SVM | Unsupervised and Supervised | *Tsafnat et al. (2011)* |
| P6 | 85 Numerical features in four categories (genomic, epigenetic, expression, network) | Detection of cancer-related long non-coding RNA | RF, NB, SVM, LR and KNN | Supervised | *Zhang et al. (2018)* |
| P8 | Z-score features, representing chromosome arm gains and losses | Detection of aneuploidy | SVM | Supervised | *Douville et al. (2018)* |
| P10 | K-mer frequencies and frequencies of certain patterns | Distinguishing endogenous retroviral LTRs from SINEs | RF | Supervised | *Ashlock & Datta (2012)* |
| P11 | Dinucleotide frequencies | Identification and clustering of RNA structure motifs | Density-based clustering | Unsupervised | *Smith et al. (2017)* |
| P12 | Sequences of nucleotides (DNA) and categorical features | Automatization of the process of extracting discriminatory features for determining functional properties of biological sequences | Evolutionary feature construction and evolutionary feature selection | Unsupervised | *Kamath, De Jong & Shehu (2014)* |
| P14 | Numerical features | Analysis of mutants | RF | Supervised | *Segal et al. (2018)* |
| P15 | Insertion sites | Identification of potential insertion sites of mobile elements | SVM | Supervised | *Rawal & Ramaswamy (2011)* |
| P16 | Numerical features | Identification of somatic LINE-1 insertions | LR | Supervised | *Tang et al. (2017)* |
| P17 | Numerical features, RNA mononucleotides, dinucleotides and trinucleotides frequencies, Fickett score | Identification of most informative features of long non-coding transcripts | 11 different feature selection approaches, SVM, RF, and NB | Supervised | *Ventola et al. (2017)* |
| P19 | Numerical and categorical features | Improve the detection and classification of TEs | NN, BN, RF, DT | Supervised | *Loureiro et al. (2013b)* |
| P21 | K-mer frequencies | Classify repetitive sequences | SVM | Supervised | *Dashti & Masoudi-Nejad (2010)* |
| P22 | Numerical features | Prediction of microRNA precursors | SVM | Supervised | *Ding, Zhou & Guan (2010)* |

(Continued)

| Publication | Data source | Task | ML algorithm | Learning method | References |
|---|---|---|---|---|---|
| P24 | Sequences of nucleotides (DNA) | Detecting repeats de novo | HMM | Supervised | *Girgis (2015)* |
| P26 | K-mer frequencies | Classify TEs using hierarchical approaches | DT, RF, NB, KNN, MLP, SVM | Supervised | *Zamith Santos et al. (2018)* |
| P27 | K-mer frequencies | Classify TEs | SVM | Supervised | *Abrusan et al. (2009)* |
| P28 | Numerical features based on structure | Identify sequence motifs conserved in each of the five major TIR superfamilies | NN, KNN, RF, and Adaboost | Supervised | *Su, Gu & Peterson (2019)* |
| P30 | Numerical features and k-mer frequencies | piRNA prediction | SVM | Supervised | *Brayet et al. (2014)* |
| P31 | Aligned genomes and binary representation (1 for mismatches and 0 for matches) | Recognition of local relationship patterns | HMM, SOM | Unsupervised | *Zamani et al. (2013)* |
| P32 | Numerical features | Compare multiple transposon insertion sequencing studies | PCA | Unsupervised | *Hubbard et al. (2019)* |
| P33 | Numerical and categorical features, nucleotide frequencies | Classify the precursors of small non-coding RNAs | RF | Supervised | *Ryvkin et al. (2014)* |
| P34 | Normalized numerical and categorical features | Prediction of transcriptional effects by intronic endogenous retroviruses | MLP NN | Supervised | *Zhang et al. (2013)* |

**Note:**
RF, Random Forest; DT, Decision Trees; BN, Bayesian Networks; NN, Neural networks; HMM, Hidden Markov Model; SVM, Support Vector Machine; NB, Naïve Bayes; LR, Logistic Regression; KNN, K-Nearest Neighbors; SOM, Self-Organizing Map; PCA, Principal Component Analysis; MLP, Multi-Layer Perceptron; FORF, first-order random forests. The full version of this table can be consulted in Table S1.

Most of the publications found in this review (Table 3) used supervised learning as the training mechanism (84%, 21 publications). We found only four publications (16%) that used unsupervised learning (Fig. 6A), which mainly addressed tasks of features selection and extraction and clustering of motif sequences.

Among the supervised learning algorithms used, RF and SVM are the most commonly found in the publications reviewed (Fig. 6B). SVM is a widely used classifier, while RF can avoid overfitting and is insensitive to noise (*Ashlock & Datta, 2012*), two features that are very useful in TE problems due to the high variability and lack of a general structure of TEs. On the other hand, we found only three publications that used Hidden Markov Models (HMM) and, in the case of *Tsafnat et al. (2011)* and *Zamani et al. (2013)*, HMMs are applied in the preprocessing step for other ML techniques. HMM is another well-known technique in bioinformatics. RED, which de novo detects repeats, is the only software found that uses HMM as the primary tool (*Girgis, 2015*). Finally, 12 of 25 publications used more than one technique to compare results and select the most optimal or to improve the accuracy obtained.

Eight publications focused on the detection or classification of TEs, and all of them used supervised techniques. RF, decision trees (DT), and SVM are the most frequent algorithms.

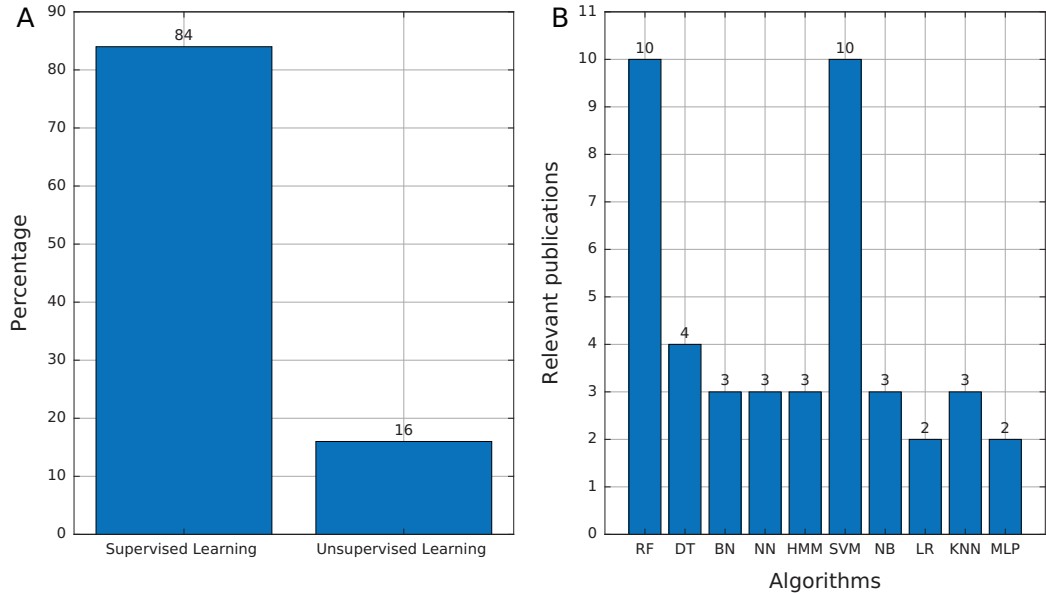

**Figure 6 Source of selected publications.** (A) Proportion of publications using supervised and unsupervised learning. (B) Supervised learning algorithms found in publications. Abbreviations: Random Forest (RF), Decision Trees (DT), Bayesian Networks (BN), Neural networks (NN), Hidden Markov Model (HMM), Support Vector Machine (SVM), Naïve Bayes (NB), Logistic Regression (LR), K-Nearest Neighbors (KNN), and Multi-Layer Perceptron (MLP).

The use of preprocessing methods to extract features from DNA or RNA (Fig. 7) was a common finding (for a review on the extraction and selection of features from biological sequences see, *Kamath, De Jong & Shehu, 2014*). For example, *Loureiro et al. (2013b)*, *Zamith Santos et al. (2018)* and *Abrusan et al. (2009)* used k-mer frequencies as features. On the contrary, *Schietgat et al. (2018)*, *Arango-López et al. (2017)* and *Loureiro et al. (2013a)* used numerical and categorical features mainly based on structures. Other purposes of ML in TEs included detecting the boundaries of mobile elements (*Tsafnat et al., 2011*), identifying insertion sites of somatic LINEs insertions (*Tang et al., 2017*), and other mobile elements at the genome level (*Rawal & Ramaswamy, 2011*), detecting aneuploidy in patients with cancer through LINEs (*Douville et al., 2018*), detecting conserved motifs in TIR elements (*Su, Gu & Peterson, 2019*), distinguishing endogenous retroviral LTRs from SINEs (*Ashlock & Datta, 2012*), and comparing multiple studies on transposon insertion sequencing (*Hubbard et al., 2019*). The last study used an unsupervised algorithm based on principal component analysis (PCA) to reduce the feature dimensions and improve the clustering analysis.

We also found publications that applied ML to other genomic data than TEs (these publications can be found in Table 2). Long non-coding RNAs (lncRNAs) are gaining attention because of critical biological functions suggested by recent studies (for a review see, *Mercer, Dinger & Mattick, 2009*). Some of the ML applications found included the detection of cancer-related lncRNA (*Zhang et al., 2018*), the discrimination of circular RNAs from other lncRNAs (*Chen et al., 2018*), and selection of the most informative features of lncRNA (*Ventola et al., 2017*). Other applications in the RNA field address the
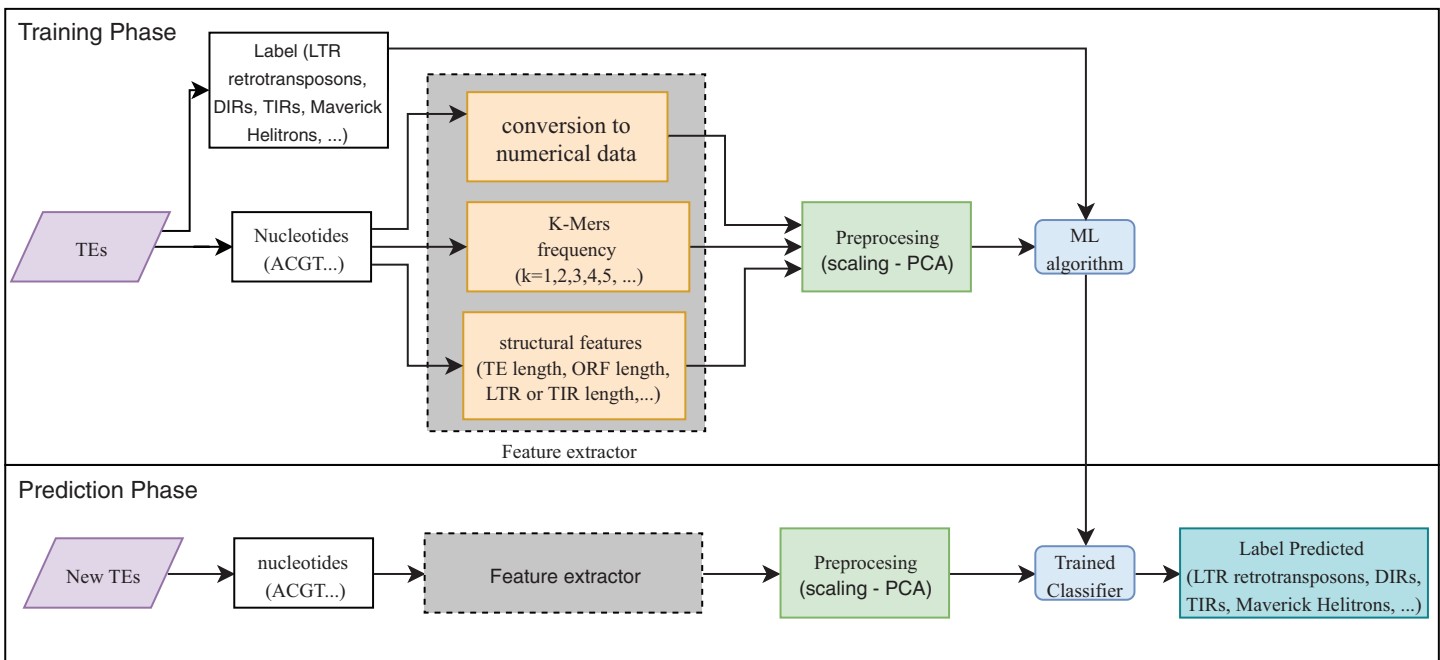

**Figure 7 Overall workflow in supervised learning ML tasks applied to TEs.**

identification and clustering of RNA structure motifs (*Smith et al., 2017*), prediction of microRNA precursors (*Ding, Zhou & Guan, 2010*), prediction of piRNA (*Brayet et al., 2014*), and classification of small non-coding RNAs (*Ryvkin et al., 2014*). Although TEs are DNA molecules, the techniques applied to lncRNA could be extrapolated to TEs since they are composed of long non-coding regions containing motifs. For example, LTR retrotransposons have two highly similar characteristic Long Terminal Repeats (LTR) that usually contain Short Inverted Repeat (SIR) motifs TG-5′ and 3′-CA at both ends (*Mascagni et al., 2015*; *Yin et al., 2013*), as well as one to three AT-rich regions with one or two TATA-boxes and a polyadenylation signal (AATAAA motif) (*Benachenhou et al., 2013*; *Gao et al., 2012*). Consequently, the approaches implemented in *Zamani et al. (2013)* can be beneficial for predicting patterns inside TEs, yielding better results in classification processes.

Regarding DL, we found five articles that addressed TE classification (*Nakano et al., 2018b*), the detection of long intergenic non-coding RNA (lincRNA) using different coding schemes and outperforming SVM results (*Yu, Yu & Pan, 2017*), the use of DL to predict enhancers based on chromatin features (*Kim et al., 2016*), and the use of CNNs to classify TEs (*Da Cruz et al., 2019*). Lastly, the fifth article reviewed the applications of DL in genomics (*Eraslan et al., 2019*).

The first ideas on DNN were discussed in the 1990s, although mature concepts on the subject matter appeared in the 2000s (*Yu, Yu & Pan, 2017*). Auto-encoders, which can perform non-linear dimensionality reduction by training a multilayer neural network with a small central layer to reconstruct high-dimensional input vectors (*Zou et al., 2018*), have been used by *Yu, Yu & Pan (2017)* with a setting of two layers. They demonstrated

**Table 4 Deep learning architectures used in genomic data reviewed in *Eraslan et al. (2019)*. Architecture details used in each work can be consulted in Table S2.**

| Dataset features | Task | DNN type | Framework or language | Year | References |
|---|---|---|---|---|---|
| Presence of binding motifs of splice factors or sequence conservation | Predict the percentage of spliced exons | Fully connected NN | TensorFlow | 2017 | Jha, Gazzara & Barash (2017) |
| Numerical features, k-mer frequencies ($k = 1, 2, 3, 4$) | Prioritize potential disease-causing genetic variants | Fully connected NN | Matlab | 2016 | Liu et al. (2016) |
| Chromatin marks, gene expression and evolutionary conservation | Predict cis-regulatory elements | Fully connected NN | Python | 2018 | Li, Shi & Wasserman (2018) |
| Microarray and sequencing data | Predict binarized in vitro and in vivo binding affinities | Convolutional NN | Python + CUDA | 2015 | Alipanahi et al. (2015) |
| A 1,000 bp sequence | Predict the presence or absence of 919 chromatin features | Convolutional NN | LUA | 2015 | Zhou & Troyanskaya (2015) |
| A 600bp sequence (one-hot matrix) | Predict 164 binarized DNA accessibility features | Convolutional NN | Torch7 | 2016 | Kelley, Snoek & Rinn (2016) |
| DNA sequence (one-hot matrix) | Classify transcription factor binding sites | Convolutional NN | Torch7 | 2018 | Wang et al. (2018) |
| DNA sequence (one-hot matrix) | Predict molecular phenotypes such as chromatin features | Convolutional NN | TensorFlow | 2018 | Kelley et al. (2018) |
| DNA sequence (one-hot matrix) and DNAse signal | DNA contact maps | Convolutional NN | Python | 2018 | Schreiber et al. (2018) |
| DNA sequence (one-hot matrix) and DNAse signal | DNA methylation | Convolutional NN | Theano + Keras | 2017 | Angermueller et al. (2017) |
| DNA sequences | Transform genomic sequences to epigenomic features | Convolutional NN | PyTorch | 2018 | Zhou et al. (2018) |
| K-mer frequencies and their positions | Predict translation efficiency | Convolutional NN | Keras | 2017 | Cuperus et al. (2017) |
| DNA sequence (one-hot matrix) and DNAse signal | Predict RNA-binding proteins | Convolutional NN | TensorFlow | 2018 | Budach & Marsico (2018) |
| Numerical features | Predict microRNA (miRNA) targets | Convolutional NN | – | 2016 | Cheng et al. (2015) |
| Numerical features | Aggregate the outputs of CNNs for predicting single-cell DNA methylation state | Recurrent NN | Theano + Keras | 2017 | Angermueller et al. (2017) |
| RNA sequence (one-hot matrix) | Predict RNA-binding proteins | Recurrent NN | Keras | 2018 | Pan et al. (2018) |
| DNA sequence (one-hot matrix) | Predict transcription factor binding and DNA accessibility | Recurrent NN | Theano + Keras | 2019 | Quang & Xie (2019) |
| RNA sequence (weight matrices) | Predict the occurrence of precursor miRNAs from the mRNA sequence | Recurrent NN | Theano + Keras | 2016 | Park et al. (2016) |
| Gene expression level (binary, over or under-expressed) | Predict binarized gene expression given the expression of other genes | Graph-convolutional NN | Torch7 | 2018 | Dutil et al. (2018) |
| Gene expression profile and protein-protein interaction network | Classify cancer subtypes | Graph-convolutional NN | – | 2017 | Rhee, Seo & Kim (2017) |
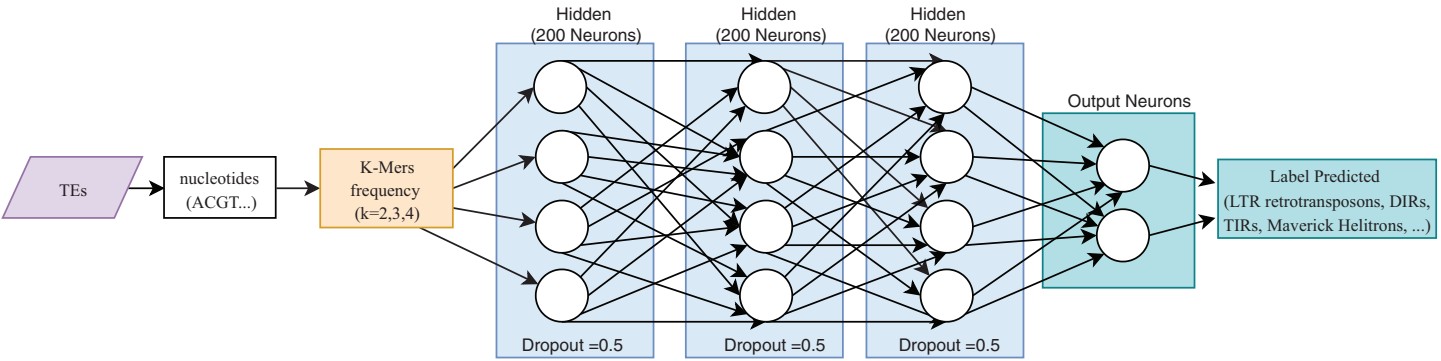

**Figure 8 Overall FNN architecture used by Nakano et al. to classify TEs.** Based on *Nakano et al. (2018b)*.

better results than SVM in the prediction of lncRNA. Moreover, other publications have reported better results from DL compared to conventional ML techniques. *Eraslan et al. (2019)* reviewed several DL architectures used in genomics (Table 4), showing improved predictive performance over ML methods, including Logistic Regression, DT or RF. Since no publications were found addressing TE detection, the application of auto-encoders can be a novel way to predict TEs with long non-coding regions (such as LTR retrotransposons).

One of the most useful characteristics of DL architectures is that DNN can automatically learn non-linear features since each layer uses multiple linear models, and the outputs are transformed by non-linear activation functions, such as sigmoid functions or rectified-linear unit (*Eraslan et al., 2019*). This process could facilitate classification tasks that include, for instance, distinguishing superfamilies of LTR retrotransposons (*Copia* and *Gypsy*). Furthermore, *Nakano et al. (2018b)* used these advantages to improve the hierarchical classification of TEs (Fig. 8) through FNN and using k-mer frequencies as features.

Advances in DL can be attributed to the use of frameworks, which facilitate the implementation of crucial operations required to build and train neural networks, such as Keras (*Chollet, 2015*), tensorFlow (*Abadi et al., 2016*), Theano (*Bergstra et al., 2011*) or Pytorch (*Paszke et al., 2017*). These operations include matrix multiplication, convolution, and automatic differentiation (*Eraslan et al., 2019*), allowing users to specify their models more easily and quickly. Another advantage is that users do not need to parallelize their codes since frameworks like tensorFlow can do it automatically.

Convolutional neural networks have been widely used in genomics. Most of the publications shown in Table 4 take advantage of the ability of CNNs to extract high-level features directly from sequences (in most cases, using one-hot codification). These features were then passed to other layers (i.e., fully connected layers) to calculate the final results. Recently, CNNs have been applied to the classification of TEs and have shown better results than conventional bioinformatics software such as PASTEC and REPCLASS. Accordingly, benefits can be gained from the use of this kind of neural networks

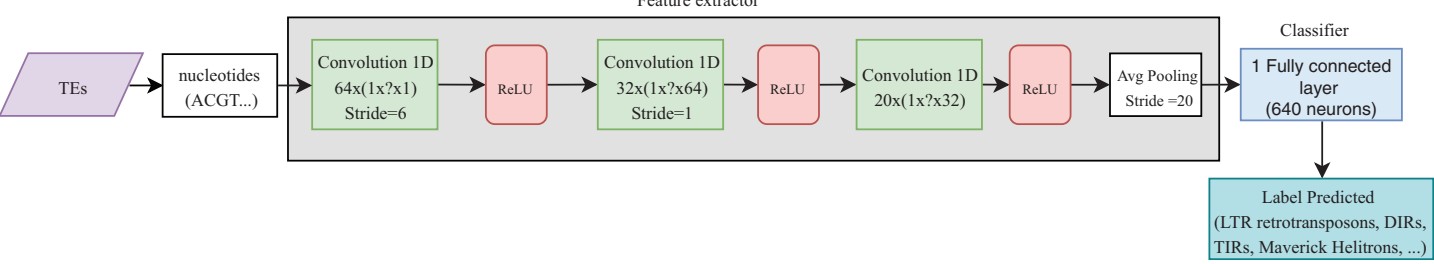

**Figure 9 Overall CNN architecture used by da Cruz et al. to classify TEs.** Based on *Da Cruz et al. (2019)*.

(*Da Cruz et al., 2019*) (Fig. 9). Since TEs display different structural features, specific motifs, and promoters, CNNs can find features that are not calculable with conventional methods. This ability can provide useful information to researchers interested in understanding the diversity and characteristics of TEs, as well as improving the detection and classification of these elements. Other architectures such as RNNs have been applied in distinct tasks in genomics, such as the prediction of binding sequences. A crucial feature of this kind of DNNs is the implementation of memory. An application of RNNs in TEs is the identification of boundaries, which, in most orders, are composed of short duplications at both ends (Target Site Duplications—TSD) and, in some orders, (i.e., LTR retrotransposons and TIR DNA transposons) of non-coding repeat regions (long terminal repeats for LTR retrotransposons and terminal inverted repeat for TIR DNA transposons). On the other hand, since nucleotides from TEs are used as input, it is likely to have more variables than individuals, leading to overfitting in the training steps. Thus, auto-encoders can be used to reduce the number of features in a non-linear way, helping to overcome this issue.

To summarize, the ML techniques already used in TEs are mainly RF, DT, and SVM. Although most publications use supervised learning, some articles can be found using unsupervised learning, mainly for extracting and selecting characteristics. Only two articles were found that applied DL (one publication used FNN and the other CNN) to the classification of TE, but they aimed to predict TE orders. Therefore, more research is needed on DL approaches.

## Parameters and metrics applied in algorithms and architectures (Q3)

To ensure that ML architectures do not exclusively learn patterns of the training dataset, there are several techniques used to split information into different datasets, such as the hold-out and k-cross-validation methods. These methods should be used in problems with information of any kind. Particularly with data of genomic origin, k-cross-validation seems to be the most popular (*Ma, Zhang & Wang, 2014*). Using $k = 10$, different studies demonstrated high accuracies for long non-coding RNAs (*Chen et al., 2018*; *Zhang et al., 2018*), for selecting features for classification of biological sequences (*Kamath, De Jong & Shehu, 2014*), analyzing insertion sites of somatic LINEs in ovarian cancer (*Tang et al., 2017*), and improving the classification of TEs (*Loureiro et al., 2013b*;

*Nakano et al., 2018b*). Meanwhile, $k = 5$ was used by *Segal et al. (2018)* to infer the importance of specific genes for growth under laboratory conditions.

In ML tasks, it is essential to have curated datasets (*Loureiro et al., 2013a*). The quality of TE and other genomic datasets is complex to evaluate, and their nature can influence the final results. Databases such as PGSB contain genomic TE sequences from many species, while repetDB or RepBase comprise consensus sequences of TEs. *Ashlock & Datta (2012)* proposed that, although consensus sequences have been used to train several ML algorithms (i.e., TEclass (*Abrusan et al., 2009*) and REPCLASS (*Feschotte et al., 2009*)), this type of datasets caused poor results. The authors also recommended taking this into account for ML projects in genomics.

A key aspect in the field of artificial intelligence is the calculation of metrics that represent the performance of the algorithms and architectures. Classification or detection tasks mostly rely on defining two classes, positive and negative. Accordingly, the predicted results are marked as true positive if they were classified as positive and are contained in the positive class, while as false negatives if they were rejected but did not belong to the negative class. Also, candidates that appear in the negative set that were classified as positives are counted as false positives, and all others are classified as true negatives (*Tsafnat et al., 2011*). Most metrics are based on the frequencies of these markers (Table 5).

The most popular metrics in ML projects are accuracy, sensitivity, specificity, precision, recall, F-score, and ROC curves (*Ma, Zhang & Wang, 2014*). However, these are not appropriate in every case, especially when the positive and negative data sets are unbalanced. For example, ROC curves are not used in TE classification, because only a small portion of the genome contains certain TE superfamilies. In this case, it is more interesting to recognize positive results than predict negative candidates correctly through precision-recall curves (PRC) (*Schietgat et al., 2018*). Also, instead of using accuracy, AUC and PRC are used for the feature construction and selection of classification of biological sequences (*Kamath, De Jong & Shehu, 2014*) and the identification of long non-coding RNA (*Ventola et al., 2017*).

In hierarchical classification problems, there is no consensus on which metrics should be used (*Zamith Santos et al., 2018*), although a set of evaluation measures have been proposed such as hierarchical Precision, hierarchical Recall, and hierarchical F-measure (Fig. 10) (*Nakano et al., 2017*). Since TE classification systems follow a hierarchical topology, these metrics can contribute to improving the measurement of algorithms and architectures to classify TEs.

In brief, because the detection and classification of TE can be covered from different approaches (binary problems, multi classes or hierarchical classification), multiple metrics can be applied. However, it is necessary to use those metrics that are not affected by unbalanced data sets, which is a problem linked to these types of data. Although in genomics, the k-cross-validation method is the most common, the k-value depends on the size of the training dataset. The articles evaluated in this review used values of $k = 5$ and 10. On the other hand, it was found that the nature of the data (genomic sequences or consensus) affects the performance of ML algorithms, so some authors recommend the use of genomic sequences.

**Table 5 Metrics used in TEs and other similar task.** Adopted from (*Kamath, De Jong & Shehu, 2014*; *Brayet et al., 2014*; *Ma, Zhang & Wang, 2014*; *Yu, Yu & Pan, 2017*; *Smith et al., 2017*; *Chen et al., 2018*; *Schietgat et al., 2018*; *Segal et al., 2018*). D for detection and C for classification.

| Metric | Representation | Observations | Tasks in which it was used |
|---|---|---|---|
| Accuracy | $\dfrac{(TP + TN)}{(TP + FP + FN + TN)}$ | Measures the percentage of samples that are correctly classified | D, C |
| Precision | $\dfrac{TP}{(TP + FP)}$ | Percentage of correct predictions | D |
| Sensitivity (recall) | $\dfrac{TP}{(TP + FN)}$ | Represents the proportion of positive samples that are correctly predicted | D, C |
| Specificity | $\dfrac{TN}{(TN + FP)}$ | Represents the proportion of negative samples that are correctly predicted | D |
| Matthews correlation coefficient | $\dfrac{TP \times TN - FN \times FP}{\sqrt{(TP + FN) \times (TN + FP) \times (TP + FP) \times (TN + FN)}}$ | It can be a key measurement because it is a balanced measurement, even if the sizes of positive and negative samples have great differences | D |
| Positive predictive value | $\dfrac{TP}{(TP + FP)}$ | Percentage of correctly classified positive samples among all positive-classified ones | D, C |
| Performance coefficient | $\dfrac{TP}{(TP + FN + FP)}$ | Ratio of correct predictions belonging to the positive class and predictions belonging to the false class | D |
| F1 score | $\dfrac{2 \times TP}{(2 \times TP + FP + FN)}$ | Harmonic mean of precision and sensitivity | D |
| Precision-recall curves | Graphics | Plots the precision of a model as a function of its recall | D, C |
| Receiver operating characteristic curves (ROCs) | Graphics | Commonly used to evaluate the discriminative power of the classification model at different thresholds | C |
| Area under the ROC curve (AUC) | Graphics | Summary measure that indicates whether prediction performance is close to random (0:5) or perfect (1:0). Also describes the sensitivity vs. the specificity of the prediction | D, C |
| Area under the Precision-Recall (auPRC) | Graphics | Measures the fraction of negatives misclassified as positives and plots the precision vs. recall ratio | D |
| False positive rate | 1–Specificity | Percentage of predictions marked as belonging to the positive class, but that are part of the negative class. | D |

$$A \quad hP = \frac{\sum_i |Z_i \cap C_i|}{\sum_i |Z_i|}$$

$$B \quad hR = \frac{\sum_i |Z_i \cap C_i|}{\sum_i |C_i|}$$

$$C \quad hF = \frac{2 * hP * hR}{hP + hR}$$

**Figure 10 Equations for hierarchical metrics. $Z_i$ and $C_i$ correspond, respectively, to the set of true and predicted classes for an instance $i$.** (A) Hierarchical precision, (B) hierarchical recall and (C) hierarchical F1-score.

**Table 6 Coding schemes for translating DNA characters in numerical representations. Adapted from (*Yu, Yu & Pan, 2017*).**

| Encoding schemes | Codebook | References |
|---|---|---|
| DAX | {'C':0, 'T':1, 'A':2, 'G':3} | Yu et al. (2015) |
| EIIP | {'C':0.1340, 'T':0.1335, 'A':0.1260, 'G':0.0806} | Nair & Sreenadhan (2006) |
| Complementary | {'C':-1, 'T':-2, 'A':2, 'G':1} | Akhtar et al. (2008) |
| Enthalpy | {'CC':0.11, 'TT':0.091, 'AA':0.091, 'GG':0.11, 'CT':0.078, 'TA':0.06, 'AG':0.078, 'CA':0.058, 'TG':0.058, 'CG':0.119, 'TC':0.056, 'AT':0.086, 'GA':0.056, 'AC':0.065, 'GT':0.065, 'GC':0.1111} | Kauer & Blöcker (2003) |
| Galois (4) | {'CC':0.0, 'CT':1.0, 'CA':2.0, 'CG':3.0, 'TC':4.0, 'TT':5.0, 'TA':6.0, 'TG':7.0, 'AC':8.0, 'AT':9.0, 'AA':1.0, 'AG':11.0, 'GC':12.0, 'GT':13.0, 'GA':14.0, 'GG':15.0} | Rosen (2006) |
| Orthogonal (one-hot) Encoding | {'A': [1, 0, 0, 0], 'C': [0, 1, 0, 0], 'T': [0, 0, 1, 0], 'G': [0, 0, 0, 1]} | Baldi et al. (2001) |

## Most used DNA coding schemes (Q4)

One of the most critical tasks in ML algorithms is correct data representation. In contrast to other datasets, DNA nucleotide sequences are human-readable characters, C, T, A, and G, so it is necessary to encode them in a machine-readable form (*Yu, Yu & Pan, 2017*). Table 6 shows coding schemes that can be applied to representing nucleotides by different approaches. Using deep neural networks, (*Yu, Yu & Pan, 2017*) demonstrated that the complementary scheme had the best performance, while the other schemes achieved similar predictive rates.

In some cases, input sequences need to be first transformed into k-mer counts (*Zamith Santos et al., 2018*). For example, for distinguishing between Endogenous Retroviral LTRs from SINEs, the dinucleotide (2-mer) "TT" appears more frequently in LTRs than in SINEs and LTRs have more TAs, TGs, As, and Gs before their first C than SINEs. These k-mer features add more information than the raw DNA sequences in the classification process (*Ashlock & Datta, 2012*). Additionally, k-mer frequencies of $k = 2, 3, 4$ have been used (*Nakano et al., 2017*, *2018a*) for TE classification through DL. Other examples that apply k-mer occurrences to sequence transformation are the prediction of DNA promoter regions, cis sites, HS sites, splice sites, among others (reviewed in *Kamath, De Jong & Shehu (2014)*).

Recently, an innovative way to convert sequences into numerical representations was proposed by *Jaiswal & Krishnamachari (2019)*. The authors considered three physicochemical properties, namely, average hydrogen bonding energy per base pair (bp), stacking energy (per bp), and solvation energy (per bp), which are calculated by taking the first di-nucleotide and then moving a sliding window, one base at a time. Accordingly, a classification task was performed using this process. Although this classification was carried out on the *Saccharomyces cerevisiae* genome, it can be extrapolated to other species with distinct types of TEs.

To summarize, the use of k-mers frequencies seems to be more common and get better results in ML algorithms. On the other hand, in DL architectures, they mainly use the one-hot coding scheme because the extraction is carried out automatically. Interestingly, the problem of TEs detection could be addressed using the physico-chemical characteristics of the di-nucleotides.

## CONCLUSIONS AND FUTURE WORK

ML is a powerful tool that can extract patterns, novel information, and relations from labeled data (supervised learning) or unlabeled data (unsupervised learning). These artificial intelligence approaches improve complex tasks and automate processes that would otherwise be done manually. Although ML and DL fields have been applied in areas such as genomics, human health, face recognition, and many others, the use of ML and DL in TEs is still limited. This is especially true for deep neural networks such as CNNs, which could provide opportunities to extract features that are undetected by conventional bioinformatics methods. Although TE detection and classification are very complex tasks because of the variability of these elements, there are databases with thousands of TE sequences that have been recently released. These databases can contribute training sets for obtaining better and generalized models to improve the accuracy and reliability of the results. TEs are associated with many aspects in humans (such as diseases) and plants (like intra and inter-species diversity, adaptation to the environment, among others). Therefore, a broader understanding of these elements can provide better knowledge of our genomes as well as about important crops. Accordingly, this can lead to faster and reliable diagnostic and clinical treatments in diseases like cancer and more resistant and productive crops. Unquestionably, ML and DL can support novel methods to detect, classify, and analyze repeated sequences. To date, there are few publications on the application of DL in TEs, so the door is open to proposing innovative methodologies and architectures.

Taking into account this systematic review of literature, we propose the following ideas as future work:

– To use Autoencoders to increase the size of the training datasets (data augmentation) on the TEs already classified and validated by the bioinformatics community, in order to obtain a better generalization of the ML and DL algorithms.

– To use sets of simultaneous classifiers (SVM, RF, DT, LR, among others) in order to generate separation frontiers of classes more adapted to the nature of this type of problem

and thus be able to increase the percentages of precision in the detection and classification of TEs.

– To train new artificial neural network architectures using transfer learning techniques from the results obtained by the neural networks proposed in the literature.

– To use techniques of selection of characteristics (RF) or reduction of dimensionality (PCA) in order to diminish the databases' complexity and to increase the percentages of precision in the detection and classification of TEs.

### Funding
Simon Orozco-Arias is supported by a Ph.D. grant from Departamento Administrativo de Ciencia, Tecnología e Innovación de Colombia (Colciencias), Convocatoria 785/2017. The authors and publication fees were supported by the Universidad Autónoma de Manizales, Manizales, Colombia under project 589-089 and Romain Guyot was supported by the LMI BIO-INCA. The funders had no role in study design, data collection and analysis, decision to publish, or preparation of the manuscript.

### Grant Disclosures
The following grant information was disclosed by the authors:
Departamento Administrativo de Ciencia, Tecnología e Innovación de Colombia (Colciencias), Convocatoria: 785/2017.
Universidad Autónoma de Manizales, Manizales, Colombia under project: 589-089.
LMI BIO-INCA.

### Competing Interests
The authors declare that they have no competing interests.

### Author Contributions
- Simon Orozco-Arias performed the experiments, analyzed the data, prepared figures and/or tables, authored or reviewed drafts of the paper, approved the final draft.
- Gustavo Isaza conceived and designed the experiments, prepared figures and/or tables, authored or reviewed drafts of the paper, approved the final draft.
- Romain Guyot conceived and designed the experiments, prepared figures and/or tables, authored or reviewed drafts of the paper, approved the final draft.
- Reinel Tabares-Soto analyzed the data, prepared figures and/or tables, authored or reviewed drafts of the paper, approved the final draft.

### Data Availability
The raw data generated is available in Table 2 (selected publications) and the Supplemental Tables.

## Supplemental Information

Supplemental information for this article can be found online at http://dx.doi.org/10.7717/peerj.8311#supplemental-information.

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
