# Peer review of "A systematic review of the application of machine learning in the detection and classification of transposable elements"

_PeerJ, doi:10.7717/peerj.8311_

## Round 0.1 · original submission · Major Revisions

· Academic Editor

Major Revisions

Your manuscript has been reviewed by three experts in the field. As you can see from their comments below, all of them feel that the organization of this manuscript is problematic. Perhaps, you should define the topic more clearly and remove rather general descriptions. Perhaps, you will need to revise the manuscript entirely. Please read their comments carefully and revise the manuscript accordingly.

Reviewer 1 ·

Basic reporting

This manuscript seems to be a report, not a review article. The authors should carefully reconsider and revise the structure of the manuscript. For example, the chapters "Search strategy" and "Selection of articles" are lengthily but contain few information. It is not difficult to shorten them without loosing any information.

Experimental design

no comment

Validity of the findings

no comment

Additional comments

Orozco-Arias et al. reported a systematic review of 35 papers/reports that detected and classified transposable elements (TEs) in genomes by applying machine learning (ML) approaches. For each study, the authors summarized addressing the following 4 points:
1. Are ML approaches for TE analyses advantageous compared to bioinformatics approaches?
2. Which ML techniques are currently used to detect and classify TEs or similar data?
3. What are the best parameters and most used metrics in algorithms and architectures to detect and classify TEs?
4. What are the most used DNA coding schemes in ML tasks?

The manuscript contains something informative to the current studies of TE identifications; however, it is not well organized and insufficient as a review article. First of all, point #1 raised by the authors seems to be the most important part in the review, but it not well answered in this review. I would like to know how ML (and DL) approaches improve the identification and classification of TEs in various species compared to the conventional (i.e. homology-based) approaches. How much species have been analyzed yet, and are there any species that are in particular suitable for ML-based search? How did they distinguish whether the search results were true-positives or false-positives? In addition to that, please consider the following comments as well.

Comment #1
In the title, the word “deep learning” should be deleted because as shown in the Figure 6 and Table 3, many (almost all) papers mentioned in the review is not based on a deep learning approach. Or if the authors include it in the title, DL-based approaches should be mentioned more in this manuscript.

Comment #2
In the Introduction, Figure 1 shows a detail classification of TEs; however, no explanations were not provided. Due to the diversity of TEs, the identification and classification of TEs are difficult. Therefore, to understand the importance of this review, the more detailed explanations of TEs are essential.

Comment #3
Figures are not easy to read. In particular, now many researchers including me believe pie charts should be avoided (please see https://www.data-to-viz.com/caveat/pie.html, for example).

Comment #4
In the Table 3, data source is not clear. For example, what is “RNA” or “DNA”? What is the differences between “k-mer frequencies” and “oligomer frequencies”?

Comment #5
References should be carefully revised.

Reviewer 2 ·

Basic reporting

no comment

Experimental design

no comment

Validity of the findings

no comment

Reviewer 3 ·

Basic reporting

The authors should clearly define what they mean with the TE tasks they are addressing. They mainly talk about TE detection and TE classification, but it would be good to properly define these tasks. Moreover, in the text they also talk about TE identification, TE prediction, TE analysis,... It is not clear whether these are considered as synonyms for detection or classification, or denote different tasks.

The formulation of the research questions should not involve methodological details (e.g. question Q2).

The structure could be improved. Some paragraphs in the response to the research questions (e.g. Q1: the paragraph starting with "In plants, TEs have been related") are more suited for the introduction.

It would be good to end each research question response with a small paragraph, summarising the answer to the research question.

Experimental design

In the Survey Methodology section, the text is well structured into the different steps, but it would be more clear for the reader if the exact same terminology as in Fig 2 is used (throughout the text and for the section titles).

I'm wondering whether there was a search criterion related to the publication date, since all included articles date from 2009 or later. Does it mean that there were no publications on this topic prior to 2009? (This would seem to contradict the fact that there is a review paper from 2010 on this topic.)

The response to research question Q2 also involves publications that apply ML to data similar to TEs. However, here the article seems to be less systematic in its review. In any case, it is not clear how the discussed publications are selected.

Validity of the findings

The results seem valid.

However, in the response to Q2, I'm not sure what an analysis of the programming language used to implement the machine learning algorithms adds to the contribution of this paper.

Also, for Q3, concepts of training/validation/test set, cross validation, and all the evaluation measures considered are not at all specific for TE detection or classification. These are concepts used in any machine learning study. Furthermore, I assume the evaluation measure will depend on the task, i.e. TE detection will require different evaluation measures than TE classification.

Additional comments

My main comments have been covered by the 3 areas above.

Some minor comments:
- The caption of Fig 6 seems to be incomplete.
- As far as I know, "neuronal networks" is not standard terminology. Please change to "neural networks".

---

## Round 0.2 · Minor Revisions

· Academic Editor

Minor Revisions

Your revised manuscript has been reviewed by the same two referees. As you can find from their comments below, both of them request additional minor revisions. Please read their comments carefully and revise the manuscript accordingly.

Reviewer 1 ·

Basic reporting

The manuscript was improved; however, I still have minor comments as follows:

On page 4: I do not know why the authors show only the first author as the research conductor in the line 155. Is it based on the guideline of systemic review of Moher et al., 2009? I think that “We” may be more suitable.

On page 5: “TEs or similar data” is ambiguous. What is similar data?

On page 7: The term “bioinformatics software” is also ambiguous. The authors may infer “bioinformatics software” as homology search-based software. The authors should carefully explain these words because it could be essential in this review.

Experimental design

no comment

Validity of the findings

no comment

Reviewer 3 ·

Basic reporting

Most of my comments have been addressed. However, I still have the following remarks, which all fall under basic reporting.

The authors removed 'deep learning' from the title, as suggested by a reviewer. This is fine, but then it should be consistently removed also from the abstract and from many places in the text where 'ML and DL' are always mentioned together.

The article contains many grammar mistakes, especially in the newly added text. To give a few examples:
- Each sub-class is then classify into orders regard to the structure features and life cycle as in the case of retrotransposons (...) and finally each order is then classify into superfamilies.
- In ML algorithms, it seems to be more common and get better results k-mers frequencies.

The second paragraph of the introduction has become unnecessarily lengthy by listing all lineages. I still believe it would be good to properly define the tasks of identificaton on the one hand and classification on the other hand. Also refer to Figure 3.

Q2, Q3 and Q4 now end with a paragraph summarizing the findings, as suggested. By starting these pragraphs with "To summarize,..." the text would be better structured and easier to follow for the reader. Also Q1 would benefit from such a paragraph.

In the section on Survey Methodology, in the sentence "We conducted an exhaustive literature review", the word "we" was replaced by the first author's name. This is not done. If the authors wish to make statements about author contributions, they should consult the journal editorial team on how to do so.

Experimental design

no comment

Validity of the findings

no comment

Additional comments

no comment

---

## Round 0.3 · accepted · Accept

· Academic Editor

Accept

Since I confirm that the raised minor points have been addressed, I am happy to inform you that I will recommend its acceptance to the journal.